# Assessment of Artificial Neural Network through Drought Indices

Smit Chetan Doshi [1,*], Mohana Sundaram Shanmugam [1] and Shatirah Akib [2,*]

1 Water Engineering and Management, School of Engineering and Technology, Asian Institute of Technology, Bangkok 12120, Thailand
2 Department of Civil Engineering, School of Architecture, Design and the Built Environment, Nottingham Trent University, Nottingham NG1 4FQ, UK
* Correspondence: smitdoshi5@gmail.com (S.C.D.); shatirah.akib@ntu.ac.uk (S.A.)

**Abstract:** Prediction of potential evapotranspiration (PET) using an artificial neural network (ANN) with a different network architecture is not uncommon. Most researchers select the optimal network using statistical indicators. However, there is still a gap to be filled in future applications in various drought indices and of assessment of location, duration, average, maximum and minimum. The objective was to compare the performance of PET computed using ANN to the Penman–Monteith technique and compare drought indices standardized precipitation index (SPI) and standardized precipitation evapotranspiration index (SPEI), using two different computed PET for the durations of 1, 3, 6, 9, and 12–months. Statistical performance of predicted PET shows an RMSE of 9.34 mm/month, RSR of 0.28, $R^2$ of 1.00, NSE of 0.92, and PBIAS of $-0.04$. Predicted PET based on ANN is lower than that the Penman–Monteith approach for maximum values and higher for minimum values. SPEI–Penman–Monteith and SPI have a monthly correlation of greater than 0.95 and similar severity categories, but SPEI is lower than SPI. The average monthly index values for SPEI prediction show that SPEI–ANN captures drought conditions with higher values than SPEI–Penman–Monteith. PET–based ANN, performs robustly in prediction, fails by a degree of severity classification to capture drought conditions when utilized.

**Keywords:** artificial neural network; drought indices; statistical analysis; evapotranspiration; United Kingdom





## 1. Introduction

According to the World Water Assessment Programme (2015), by 2050, global food demand is predicted to rise by 70%. Thus, an increase in agricultural activities would be required, leading to 70% of total freshwater withdrawal worldwide [1,2]. Sustainability in water usage would be demanded with the increasing population and finite water supply [3]. One of the most pressing issues faced by the world in the 21st century is the lack of access to water [4]. The severity of water deficits on the world's surface is reflected in various drought indices. Droughts are unavoidable and detrimental to the ecological and social system. Economically, drought impacts are severe across the globe, especially meteorological droughts, requiring thorough investigation [5,6]. Further–more, climate change and socioeconomic development have significantly altered the regional supply, and moisture demand, and it is essential for risk management to comprehend the dynamics of drought and the effects of shifting supply and demand [7,8]. Potential Evapotranspiration (PET) is a components that significantly affects drought conditions [9]. For effective water management, a precise estimate of evapotranspiration (ET) is required, and deep learning algorithms could greatly benefit forecasting [10].

Next to precipitation comes the most significant component, ET in the hydrologic budget. The spatial variance of ET could be visualized both regionally and seasonally, and

during drought according to weather [11]. ET is a combination of two processes, evaporation and transpiration [12–14]. The number of water sources, soil moisture, and vegetation growth are significantly impacted by ET [15]. Therefore, it is critical to compute ET. The ET can be measured on the field using a lysimeter through a water balance approach or inferred by an empirical approach using climatological data. On the field, using a lysimeter would not always be possible due to time constraints and maintaining unhindered measurements of experiments [16]. The data requirements for empirical methods of estimating PET ranges from air temperature to radiation, wind speed, precipitation, and relative humidity [17]. Due to climatic data constraints or that they were specially developed for a given climate region, PET's reliability varies as methodology varies [18–21]. Due to a lack of understanding of the ET process physically and the absences of all pertinent data, the ET has added inaccuracy [16]. The Penman–Monteith method superior to estimating PET compared to other empirical methods for various climatic circumstances [22–26]; however, the daily time–scale is a challenge due to the non–availability of climatic data across the study area. Accuracy of PET estimation depends on the availability of high–quality meteorological data. It is susceptible to modifications in inaccuracies of climatic parameters [27,28].

Drought is one of the costliest natural environmental disasters and an extreme event [29] impacting increasing wildfires, water scarcity, crop loss and health effects [30–36]. With no universal definition of droughts [37], it could be generally coincide where the precipitation, soil moisture, water sources and supplies are scarce [38]. Drought features based on raw data have less considerable benefits than drought indices, which are single numerical values computed from multiple climatic variables [39]. Climatic variables, precipitation, runoff, soil moisture and PET, significantly impact drought [29]. Future effects of drought are more likely to worsen under the global warming scenario [40,41]. To measure drought severity, duration and extent of drought indices have been created with precipitation or a combination of meteorological variables [29,42,43]. Standardized Precipitation Index (SPI) and Standardized Precipitation Evapotranspiration Index (SPEI) are the most frequently utilized indices for meteorological drought evaluation [44–49]. SPI uses monthly precipitation totals, and SPEI adds to that with PET, which is where the difference in result lies [50].

ET is a complicated nonlinear phenomenon as it depends on various climatic factors, crop characteristics and growing cycles [51]. Due to the unavailability of climatic data, selecting a methods to capture the ground reality is challenging [52]. Current challenges in ET determination include defining and understanding, measuring in–situ, parameterizing and estimating remotely at catchment scale ET [53]. The artificial neural network (ANN) is a tool for assessing PET [52]. In numerous fields of knowledge, ANN has been successfully employed to model the relationship involving complex transient series [52]. The ANN allows capturing of more complex properties of the data, which are sometimes challenging to do when using traditional statistical methods. It does not require in–depth details about the physical processes as they are explicitly stated as input–output models with a mathematical form [54]. Numerous scholars have used ANN in their studies to estimate ET as a function of meteorological factors, further discovering acceptable and even superior outcomes to the usual, conventional approach [51,54–57].

The recent literature review includes the estimation of PET through several approaches, such as comparison of empirical equations [58–61] or prediction through complex machine learning algorithms [62–70] either through remotely sensed data or through meteorological observed station data. Drought assessments with various PET methods and prediction of drought indices through machine learning algorithms have been carried out in recent research studies [9,71–74]. The research generally addresses which empirical method or machine learning algorithm is robust in determining PET, assessing various drought indices, and predicting and forecasting PET or drought indices. A study has been carried out for the United Kingdom (UK), which indicates an increase in river flow over the last five decades (before 2015) [75], potential evaporation from 1961–2012 [76], precipitation [76], and ET [77]. Major drought events experienced in the UK are 1798–1808, 1854–1860, 1887–1888, 1890–1909, 1921–1922, 1933–1934, 1959, 1976, 1990–1992, and 1995–1997, with

the main cause being the long duration of dry weather [78]. The UK has suffered clusters of drought in the form of long and short drought series [79,80]. The following conclusions were drawn with the recent study carried out for the UK indicating projected changes in droughts using SPEI and SPI: increase in drought risk with increasing global mean surface temperature, drought frequency seasonality, duration greatly varies spatially, increase in English regions and Wales, decreases in North and West Scotland, and a large difference between SPI and SPEI was observed [81]. SPEI–6 was the best predictor of drought impacts on agriculture in UK regions compared to the reported drought impacts, however, the findings indicate that the relationship varies spatially between the large heterogeneous regions and needs smaller spatial units [82]. With the review, we realized that there is minimal work addressing the application of the output of artificial neural network–based PET to drought indices. The research gap we are addressing covers this aspect with a specific case study of the UK, considering the highest resolution observed data as the input data source.

The objective of the study is to investigate the performance of predicted PET based on ANN in comparison to Penman–Monteith using drought indices for the UK region. We selected meteorological drought indices SPEI and SPI and compared the results statistically. The drought indices were choosen to analyze the drought phenomenon for the given study area based on its widely used approach [83–86]. The highlight is the approach method, not only predicts PET but also tests its performance when used as drought indices with various cumulative periods (1, 3, 6, 9, and 12–months) as well as the highest resolution of observed data utilized for the whole country at a monthly scale. The main goals were:

- Computation of PET and drought indices (SPEI and SPI) using high resolution gridded data.
- ANN model development and prediction of PET.
- Computation and comparison of ANN–based PET with the observed data–based PET for various drought indices.

## 2. Materials and Methods

### 2.1. Data Acquisition

The study area focused on here is the UK which is ~244,820 km$^2$ (94,530 mi$^2$), comprising England, Scotland, Wales and Northern Ireland. This study uses the package from the R platform "SPEI" (https://cran.r--project.org/package=SPEI) to compute PET using the Penman–Monteith method, SPI and SPEI. The input data required to compute PET are elevation, maximum and minimum temperature, precipitation, relative humidity, sunshine hours, sea level pressure and wind speed. The HadUK–Grid c1.1.0.0 dataset was used to obtain the gridded climate variables [87]. This dataset is at a monthly timescale of the period 1969–2021 (53 years); with an extent of 48.83° N–60.86° N × 12.61° W–4.59° E providing complete coverage across the UK with a resolution of 1 km × 1 km [87]. The gridded data set is derived from land surface observation of the UK network with further interpolation from meteorological station data [87]. The elevation dataset is obtained from the GTOPO30 global digital elevation model (DEM) with a horizontal grid spacing of ~1 km [88]. Averaged yearly spatial distribution of weather data across the UK using monthly weather data and elevation are showcased (Figure 1).

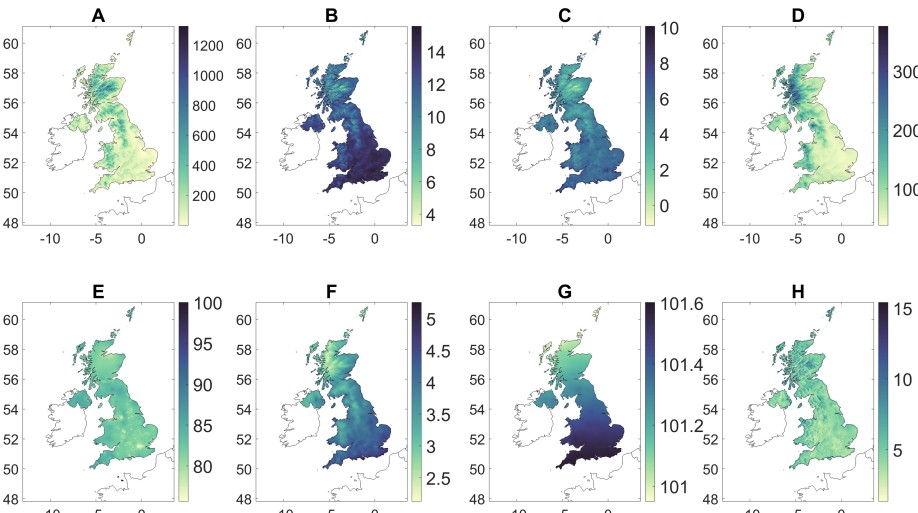

**Figure 1.** Average yearly spatial distribution of climatic data and elevation: (**A**) elevation (m), (**B**) average air maximum temperature (°C), (**C**) average air minimum temperature (°C), (**D**) total average monthly cumulative precipitation (mm), (**E**) average relative humidity (%), (**F**) average daily sunshine hours (h), (**G**) average sea level pressure (kPa) and (**H**) average wind speed (m/s).

### 2.2. Methodology

The first phase entails choosing a PET technique based on available input data and the empirical equation that may accurately reflect the actual conditions. The Penman–Monteith approach is chosen on consideration of factors. According to the FAO manual [13], the following Equation (1) is used to calculate the PET:

$$\mathrm{PET} = \frac{0.408\Delta\,(\mathrm{R_n} - \mathrm{G}) + \gamma\frac{900}{\mathrm{T}+273}\mathrm{u_2}(\mathrm{e_s} - \mathrm{e_a})}{\Delta + \gamma\,(1 + 0.34\mathrm{u_2})} \tag{1}$$

where, PET is potential evapotranspiration in mm/day, $R_n$ is net radiation at crop surface in MJ/m$^2$ day, G is soil heat flux density in MJ/m$^2$ day, T is mean daily air temperature at 2 m height in °C, $u_2$ is wind speed at 2 m height in m/s, $e_s$ and $e_a$ are saturation and actual vapor pressure in kPa (the difference would result in vapor pressure deficit), $\Delta$ is the slope vapor pressure curve in kPa/°C, and $\gamma$ is the psychometric constant (kPa/°C). It has been assumed that the reference crop will be short for the entire UK coverage, which could increase some of the uncertainty. The local and regional management structure in the UK region, which may contain an abundance or scarcity of water supplies, are additional elements that could impact uncertainty.

Once the PET has been determined using the Penman–Monteith equation, the ANN model is created. The ANN model is a multilayer feed forward backpropagation neural network, commonly referred to as a multilayer perceptron (MLP). There are three layers in a neural network: input, hidden and output layer. In contrast to the Penman–Monteith based PET, the performance of the output determines the type of input parameter, training algorithm, activation functions, neurons and hidden layers [89]. To test the performance various statistical indicators like Root Mean Square Error (RMSE), RMSE–observations standard deviation ratio (RSR), R–squared coefficient ($R^2$), Nash Sutcliffe model Efficiency coefficient (NSE) and Percent Bias (PBIAS) are used. It adheres to the first–order gradient slope with a steep descent approach. The network propagates errors, which are the discrepancy between output PET and the empirical–based PET for a given model set. This procedure is repeated until the required error tolerance is attained [90,91]. Weight and bias are adjusted to optimize the training of the Levenberg–Marquardt algorithm. The ANN is trained with this technique as it is the most effective training approach. Before prediction is carried out, training is conducted for the given set of input and output values. The output value is constrained to a certain finite value based on the selection of the activation

function. Learning parameters such as learning rate, momentum value, error function, epoch size and gain of transfer function affect the network performance [92]. Initial weights are random, but over iterations of the learning algorithms modifies them. Equation (2) is the approach behind the ANN model.

$$Y_N = f_o[\, B' + \sum_{j=1}^{n_h} W_j'' f_h(\sum_{i=1}^{n_v} W_{ij}' X_i + B)] \tag{2}$$

Here, $X_i$ is input data, $Y_N$ is output data, $n_h$ and $n_v$ are the number of neurons in hidden and input layer, $B'$ and $B$ are the biases for output and hidden layer, $f_o$ is the activation function for output layer, $W_{ij}'$ weights connecting input and hidden layers, $f_h$ is the activation function transferring from input layer, $W_j''$ weights connecting output and hidden layer.

Figure 2 describes the network architecture used to predict the PET using a feed forward backpropagation neural network.

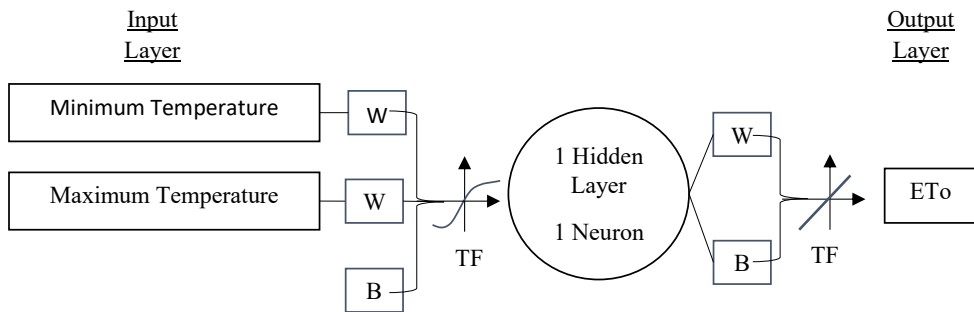

**Figure 2.** Artificial Neural Network used for training, validation and prediction (W, B and TF mean weight, bias and transfer function, respectively).

Monthly temperature data for the maximum and minimum for 53 years (1969–2021) were used as predictors. The predictand was based on the PET produced using the Penman–Monteith method. The choice of the input data was made based on its simplicity of accessibility across many regions where a similar approach was used, as well as an alternative to the lack of climatic data in the study area. The ANN model was developed grid by grid with the weights and bias initially as a random value with 1 hidden layer and 1 neuron. A total of 1000 iterations were performed, with a learning rate of 0.1. The hidden layer uses a hyperbolic tangent sigmoid transfer function, whereas the output layer uses a linear transfer function. Prior to making a prediction, it is crucial to train and validate the model using the provided predictor and predictand data. To do this, the training set was separated into the years 1969–2011 and the validation set into the years 2012–2016. Once the statistical indicators were computed, and the model was robust enough to be utilized for future predictions, the prediction dataset years 2017–2021 were tested out. The descriptive statistics in terms of training, validation and prediction of the maximum and minimum temperature predictor data used for ANN model development are showcased in Table 1.

SPI value derived from the observation record, represents the frequency of the recorded precipitation amount in the corresponding month. It assigns a classification to the precipitation amounts of a given frequency in relation to the amount for the same month over all the measurement years. A standard normal centered at zero is created from the cumulative precipitation total for the entire record. The probability distribution function used for the SPI is gamma, SPEI is log–logistic, and the method used for computing the distribution function parameters is the unbiased sample probability–weighted moments. In a similar approach to SPI, SPEI is computed, but instead of utilizing precipitation total, cumulative water balance is used. The water balance is determined using the difference between precipitation and PET. The PET in this determination differs from the previous

steps, which is Penman–Monteith based PET and ANN–based PET. The condition of the study area is assessed with the values of SPI and SPEI for various frequencies such as 1, 3, 6, 9, and 12–months [93].

**Table 1.** Descriptive statistics of the input data used for ANN model development.

|  | Minimum | Maximum | Mean | Standard Deviation | Coefficient of Variation |
|---|---|---|---|---|---|
| Maximum Temperature (°C) | | | | | |
| Training (1969–2011) | −6.21 | 28.28 | 12.51 | 0.13 | 0.01 |
| Validation (2012–2016) | −4.82 | 26.66 | 13.00 | 0.13 | 0.01 |
| Prediction (2017–2021) | −3.97 | 28.09 | 13.31 | 0.13 | 0.01 |
| Minimum Temperature (°C) | | | | | |
| Training (1969–2011) | −10.99 | 17.40 | 5.37 | 0.05 | 0.01 |
| Validation (2012–2016) | −8.06 | 16.43 | 5.74 | 0.05 | 0.01 |
| Prediction (2017–2021) | −7.14 | 17.19 | 6.00 | 0.05 | 0.01 |

## 3. Results and Discussion

The highest maximum and lowest minimum temperatures were recorded in July (6 °C to 28 °C) and February (−10 °C to 9 °C), respectively. Maximum and minimum precipitation totals were recorded in December (1500 mm/month) and May (700 mm/month), while the average relative humidity ranges from 78% to 88%. The UK experiences its highest and lowest levels of sunshine in May (188 h per month) and December (40 h per month). The average wind speed in the UK is 4.84 m/s (4.07 m/s in August and 5.61 m/s in January). As supplementary information, each meteorological dataset's monthly spatial visualization is displayed to highlight the dispersion of the data. The results are showcased with the initial comparison of PET based on the Penman–Monteith Equation and the ANN model, followed by utilizing each of the PET to compute drought indices SPEI for durations of 1, 3, 6, 9, and 12–months. It also compares the two drought indices: SPI and SPEI.

### 3.1. Penman–Monteith and ANN–Based PET

Based on the monthly weather data available for the UK study area, monthly PET values were calculated using an empirical equation and ANN model. The time series of the comparison of PET using two approaches, Penman–Monteith and ANN model, is showcased in Figure 3. The computed values are averaged over months, showcased spatially in Figure 4.

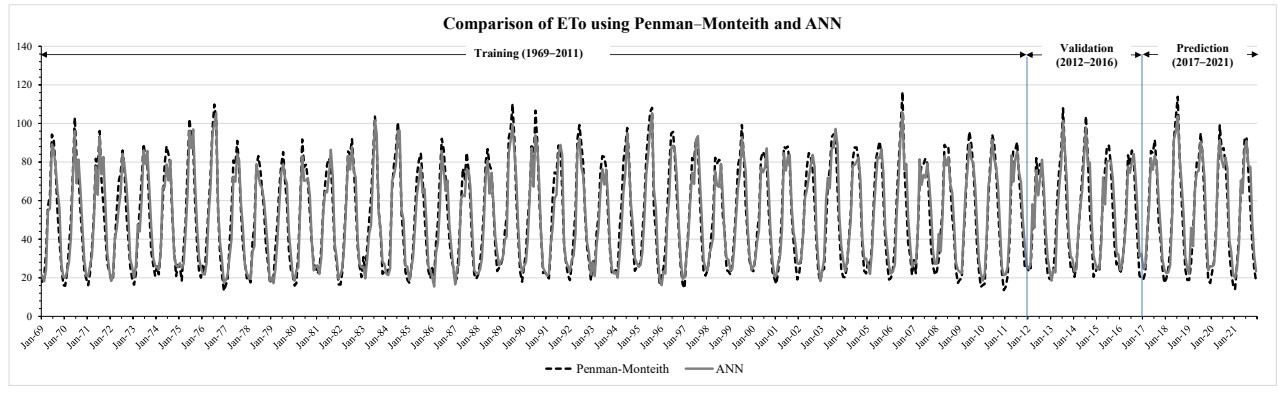

**Figure 3.** Time Series Comparison of PET using Penman–Monteith and ANN.

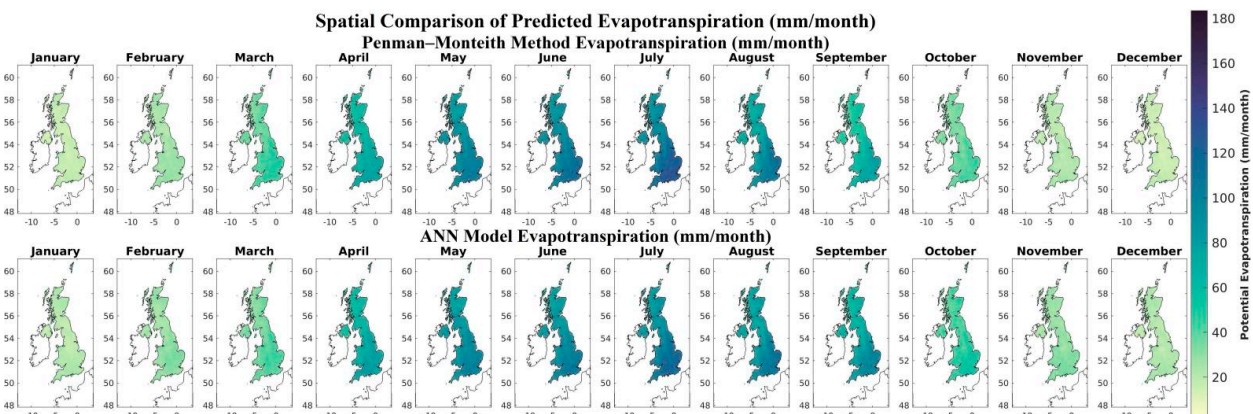

**Figure 4.** Monthly spatial visualization of PET using Penman–Monteith and ANN (prediction performance).

The analysis shows that the average PET over the past 53 years has been between 20 and 91 mm/month between December and July. The UK's highest maximum PET was recorded in July of 2006 (about 116 mm/month), and its lowest minimum was registered in December of 1976 (about 13 mm/month). When the UK is shown on an average monthly basis over 53 years, the southern part experiences more PET than the northern part. The greatest PET, 690 mm/year, was recorded in 1995. The average annual PET is 620 mm/year. The average PET for January was ~21 mm/month, February was ~24 mm/month, March was ~40 mm/month, April was ~58 mm/month, May was ~79 mm/month, June was ~86 mm/month, July was ~91 mm/month, August was ~80 mm/month, September was ~57 mm/month, October was ~38 mm/month, November was ~25 mm/month, and December was ~20 mm/month. Northolt (~71 mm/month) and Shetland (~37 mm/month) have the average monthly maximum and minimum for all the cumulative years' PET, respectively.

The performance using statistical indicators for the training, validation and prediction datasets shows, respectively, RMSE values in mm/month of 9.40, 10.72, and 9.34; RSR values of 0.30, 0.35, and 0.28; R2 values of 0.97, 0.99, and 1.00; NSE values of 0.91, 0.88, and 0.92; and PBIAS values of 0.01, −0.03, and −0.04. The training and validation indicate there is no overestimation, and hence the model is robust to be used for prediction. Further, the statistical indicators also include a satisfactory performance for prediction. The time series comparison from Figure 3 and spatial visualization from Figure 4 also depict an overlay of similarity in contrast. The Penman–Monteith method predicts the highest PET at Hayes with ~184 mm/month during July 2018 and the lowest PET at Fort William with ~1.5 mm/month during January 2021, while the ANN method predicts the highest PET at Hayes with ~161 mm/month during July 2018 and the lowest PET at Ballachulish with ~6 mm/month during January 2021. Using Penman–Monteith, the average PET across the UK is maximum in July with ~96 mm/month (maximum PET in July 2018 ~114 mm/month) and minimum in December with ~18 mm/month (minimum PET in January 2021 ~14 mm/month), whereas using ANN, the average PET is maximum in July with ~89 mm/month (maximum PET in July 2018 ~103 mm/month) and minimum in January with ~23 mm/month (minimum PET at January 2021 ~20 mm/month). When comparing PET predictions, the error difference for each month ranges from −9 mm/month to 15 mm/month. Previous studies have concluded that ANN modeling is an alternative for FAO–56 Penman–Monteith PET with a high model efficiency [16,51,94]. Compared to various other algorithms, Levenberg–Marquardt simulated the best estimate of PET [94–96]. The similar network architecture of ANN shows not only a good robust output over a general average value but also with minimal error in capturing maximum and minimum values for a trained, validated and predicted dataset across the UK.

*3.2. SPI and SPEI with Penman–Monteith and ANN–Based PET*

For the gridded data across the UK, the 1, 3, 6, 9, and 12–months values of SPI and SPEI were calculated. SPEI was computed with two different PET, one derived from Penman–Monteith and other from ANN model. Figure 5 gives a visualization of the comparison of the monthly drought indices over 53 years (1969–2021).

Average monthly results over 53 years in the UK show that the SPI and SPEI (Penman–Monteith) drought indices with durations of 1 month were during the months of April–July and October–December, 3 months during October–February (also similar SPI and SPEI), 6 months during August–February (also similar SPI and SPEI), 9 months during July–January and September–May, and 12 months during August–January and October–May. The highest moist conditions have been recorded based on SPI and SPEI for 1, 3, 6, 9, and 12–months: February 2020, July 2007, September 2012, July 2007 and August 2007. Values of SPI during that period were ~1.83, ~1.86, ~1.70, ~1.66, and ~1.69, and values for SPEI were ~1.75, ~1.76, ~1.58, ~1.56, and ~1.56. The highest drought conditions observed based on SPI and SPEI for 1, 3, 6, 9, and 12–months were during April 1974 and August 1995, October 1972, and August 1976, August 1984 and August 1995, March 1973 and October 2003, and August 1976. Values of SPI during that period were $\sim-2.46$, $\sim-2.185$, $\sim-1.85$, $\sim-2.06$, and $\sim-1.97$, and for SPEI were $\sim-1.95$, $\sim-1.91$, $\sim-1.60$, $\sim-1.82$, and $\sim-1.78$. The yearly averages showcases an increasing trend in drought index values. The extreme moist condition based on SPI for frequencies of 1, 3, 6, 9, and 12–months were observed at Moray (~4.35—September 1995), Rushbury (~4.58—July 2007), Hexham (~4.36—April 2016), Llangoed (~4.32—July 2016), and Keswick (~4.10—September 2016), whereas for SPEI, it was observed at Upton (~3.55—August 2004), Devon (~3.78—August 2012), Colyton (~4.32—August 2012), Wistow (~4.12—September 2012), and Forsinard (~3.79—December 1969). The percentage differences of extreme moist values between SPI and SPEI for 1, 3, 6, 9, and 12–months were ~22%, ~21%, ~1%, ~5%, and ~8%. The extreme drought condition based on SPI for frequencies of 1, 3, 6, 9 and 12 months were observed at Isle of Lewis ($\sim-12.99$—April 1974), Larnencekirk ($\sim-9.77$—August 1994), Ventnor ($\sim-5.69$—July 1976), Selsey ($\sim-6.35$—July 1976), and Housay ($\sim-6.41$—December 1970) whereas for SPEI were observed at Hull ($\sim-6.94$—July 1976), Aberdeenshire ($\sim-3.85$—May 2020), Bolton ($\sim-9.75$—March 1996), Clitheroe ($\sim-7.67$—June 1996), and Chorley ($\sim-5.58$—July 1996). The percentage difference of extreme drought values between SPI and SPEI for 1, 3, 6, 9 and 12 months are ~87%, ~154%, $\sim-41\%$, ~17%, and ~15%. A Comparison of SPI and SPEI has been made in the past, it indicates that the SPEI is more accurate as it incorporates PET, and it is advisable to use SPEI [49,97,98]. The results indicate that the captured monthly maximum values of SPI and SPEI are similar during periods of moist conditions but differ during periods of drought conditions except for the 12–month frequency. Moreover, under both conditions, the computed output from SPI is higher than that of SPEI, which is similar to the previous analysis in different regions [99]. The correlation coefficient between SPEI and SPI for frequency of 1, 3, 6, 9, and 12–months is > 0.95 across each month. A high correlation between SPEI and SPI has already been showcased in previous research [100–102], and the condition remains valid for the UK too. Additionally, evaluation was done for the predicted PET when utilizing ANN to compute the SPEI—1, 3, 6, 9, and 12–months drought indices. Spatial visualization is shown in Figure 6 compared to the empirically based Penman–Monteith generated PET drought indices.

The visualization reveals that, on average the predicted PET, when applied to computing drought indices does not differ greatly in comparison. However, each month is different in terms of visualization patterns during prediction, so seasonality is not integrated into the computation of drought indices. A comparison of SPEI for predictions based on a duration of 1–month indicates that January, April, May, and November have major areas under drought conditions whereas February, March, June, July, August, September, October, and December have major areas under wet conditions. For a duration of 3 months, the drought conditions mostly prevailed in the months of April, May, June, and

July, and the rest of the months present major areas under wet conditions. For a duration of 6 months, the drought conditions mostly prevailed in April, May, June, July, August, and September, and the other months present major areas with wet conditions. For a duration of 9 months, the drought conditions mostly prevailed in May, June, July, August, and September, and the other months present major areas with wet conditions. For SPEI–12, it cannot be visualized in a similar way as across all the months for drought and moist conditions. Predicted PET from the ANN used for computing SPEI for 1, 3, 6, 9, and 12–months indicates extreme wet conditions with index values of ~2.84 (January 2021), ~2.80 (February 2021), ~3.04 (February 2018), ~3.28 (April 2018) and ~3.11 (December 2020) whereas Penman–Monteith indicates ~2.91 (January 2021), ~2.86 (February 2021), ~3.01 (October 2019), ~3.22 (April 2018) and ~3.18 (June 2021). For ANN–based extreme drought conditions with index values of ~−2.92 (May 2020), ~−3.06 (June 2018), ~−2.87 (February 2017), ~−3.19 (April 2017) and ~−2.83 (June 2017) whereas Penman–Monteith indicates ~−6.21 (May 2020), ~−3.85 (May 2020), −3.10 (May 2020), ~−3.46 (May 2017) and ~−2.58 (June 2017). The differences between drought index values capturing extreme wet and drought conditions for SPEI of 1, 3, 6, 9, and 12–months using both approaches are ~0.07, ~0.06, ~−0.03, ~−0.06, and ~0.07 and ~−3.29, ~−0.79, ~−0.23, ~−0.27, and ~0.26. Average SPEIs during 1, 3, 6, 9, and 12–months frequency using Penman–Monteith across the UK when the prediction is maximum are ~1.75 (February 2020), ~1.14 (February 2020), ~1.12 (February 2020), ~1.40 (February 2020), and ~1.29 (February 2020). When it is minimum, they are ~−1.36 (April 2020), ~−1.80 (July 2018), ~−1.24 (July 2018), ~−1.31 (May 2017), and ~−1.25 (April 2019). Using ANN when the prediction is maximum, the predictions are ~1.77 (February 2020), ~1.08 (February 2020), ~1.07 (February 2020), ~1.33 (February 2020), and ~1.20 (February 2020) and when they are minimum, they are ~−1.44 (April 2020), ~−1.86 (July 2018), ~−1.34 (February 2017), ~−1.51 (May 2017), and ~−1.38 (April 2019). Previous research has focused towards predicting or forecasting drought indices [103–107] or PET [108–110] using complex approaches. The result highlight the application of predicted PET to compute drought indices SPEI of various durations. The results indicate that even though the PET of the ANN and empirical approach presented satisfactory performance through the statistical indicators, for the drought–severity classification [111] it failed for the frequency of 1–month of SPEI.

### Comparison of SPI and SPEI using ETo of Penman-Monteith and ANN

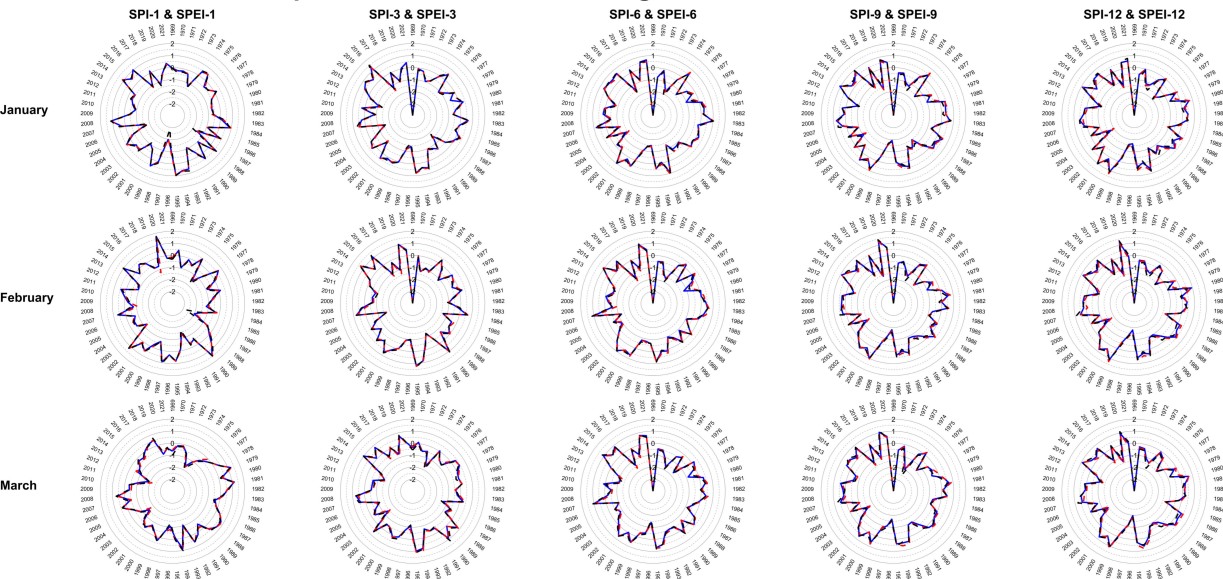

**Figure 5.** *Cont.*

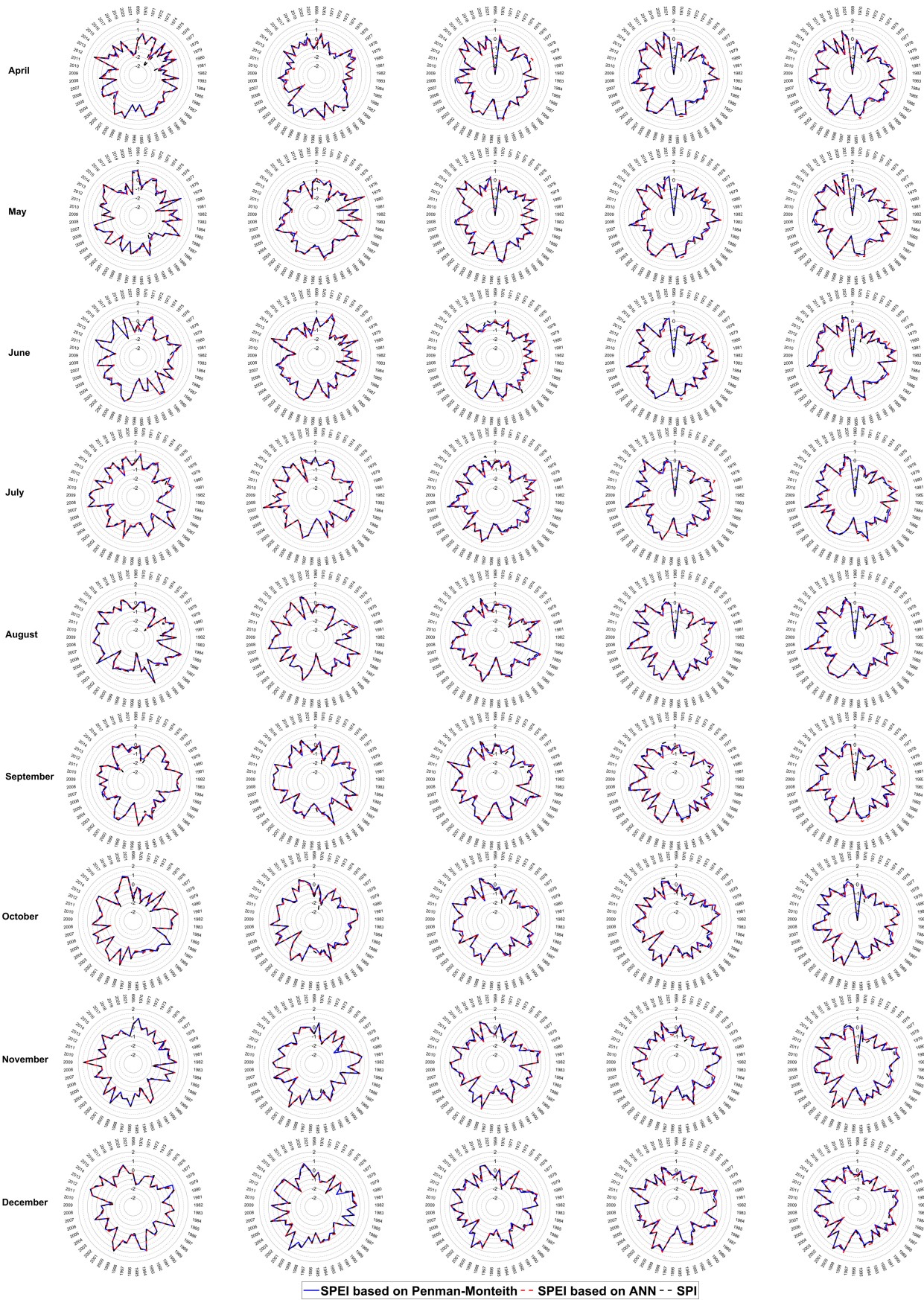

**Figure 5.** Comparison of SPI and SPEI using PET of Penman–Monteith and ANN.

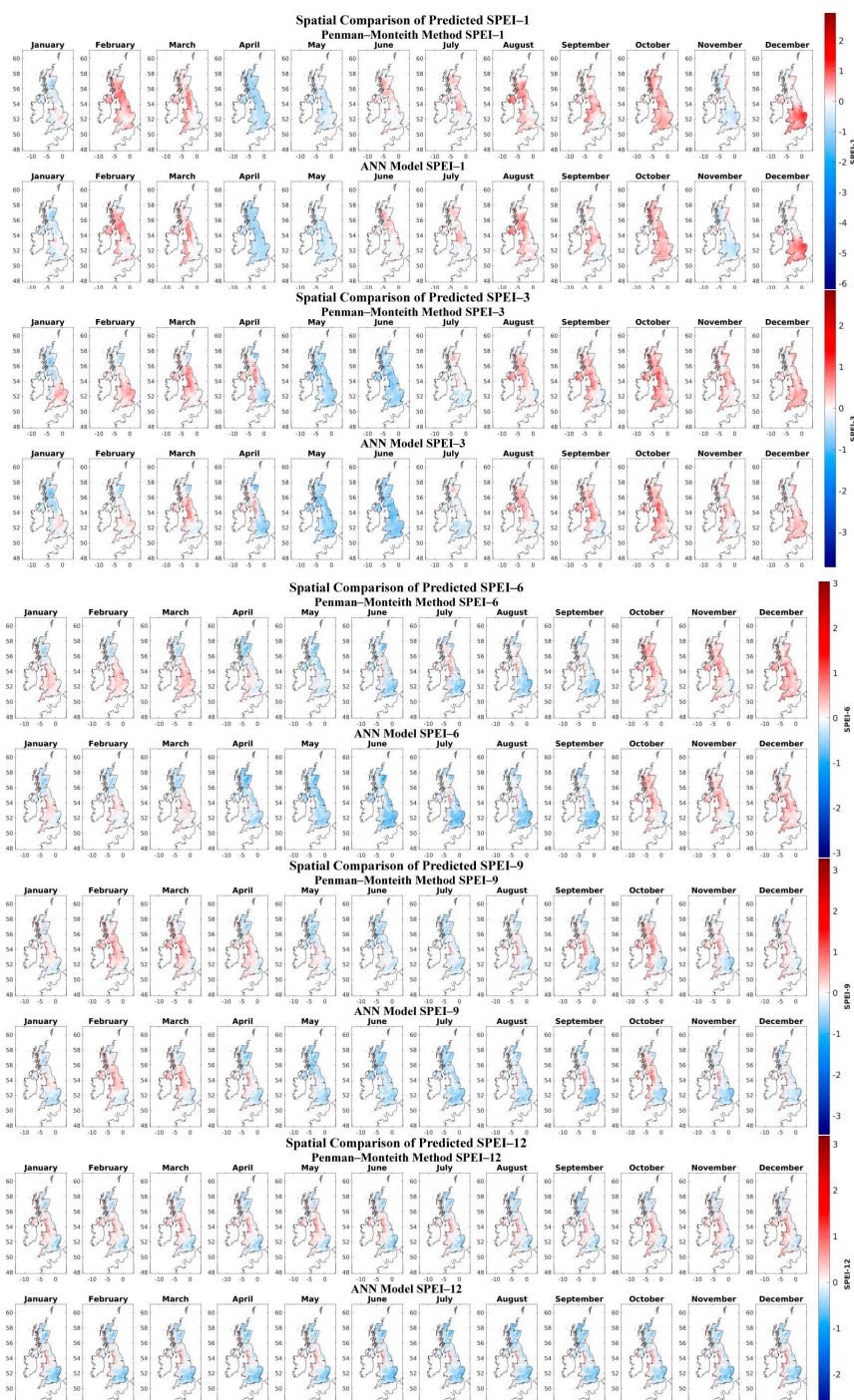

**Figure 6.** Monthly spatial visualization of SPEI (1, 3, 6, 9 and 12) using Penman–Monteith and ANN (prediction performance) (red indicates wet condition, and blue indicates drought condition).

## 4. Conclusions

This study takes into account the application of predicted PET using an ANN in terms of drought indices SPEI by comparison with empirical PET using Penman–Monteith in terms of location, duration, average, maximum and minimum.

PET, was predicted using the ANN model based on the minimum and maximum air temperatures as predictor and Penman–Monteith–based PET as predictand. A feedforward backpropagation neural network with a layer of neurons and a hidden layer was optimized with the given set of predictor and predictand through training and validation. A total of 53 years of high resolution gridded monthly data for the study region across the UK was

utilized in this study with ANN model development grid–by–grid. The performance of ANN models using time–series, spatial visualization, and statistical indicators revealed that the model could be used to predict PET. Prediction on average indicates a similar month (July) during which maximum PET is reached. However, there is one month gain (instead of December, it shows January) to the minimum PET. The differences between the averages of all months during prediction are negligible in comparison. The ANN shows a lower value than that of Penman–Monteith for the maximum PET at any given location and a higher value during minimum PET; however, the location and time are similar.

Comparison of SPI and SPEI indicates similarity between the minimum and maximum average values for durations of 3 and 6 months, but for other durations (1, 9 and 12 months), the lag is high. Averaged monthly highest moist conditions observed for SPEI and SPI are similar for all durations, and error differences are negligible, indicating that all of them present similar conditions—severely wet. Averaged monthly highest drought conditions observed for SPEI and SPI are similar only during the 12–month duration. Moreover, SPEI tends to showcase lower values in comparison with SPI for severe wet and drought conditions. Based on the drought indices computing individual maximum values across the UK, the error percentages of SPEI and SPI are higher for drought prone zones compared to wet prone zones. This indicates that on average the comparison looks similar over months, but interchangeability of drought indices still lacks if the maximum and minimum values are focused. However, all the values computed do fall in the same range of severity.

Based on the predicted ranges of PET (2016–2021), two different SPEIs are computed which showcase the application of ANN through drought indices. After visualization, it could be stated that all the regions of the UK face challenges in terms of drought and wet conditions across various months. However, the degree of extremes might vary along the months. There is a lag observed in ANN–based SPEI in capturing the extreme moist and drought conditions. For SPEI–1 the difference is relatively smaller in moist–condition averages but higher in terms of severity classification for drought conditions. For the rest of the SPEI durations, it is very small for change in classification of severity [111]. On average, in all predicted years across the UK, the drought index values for extreme drought conditions for SPEI were higher for ANN– than Penman–Monteith–based PET.

To conclude by covering maximum and minimum values, the key findings are:

- ANN–based PET shows a lower values than Penman–Monteith–based PET for maximum values and a higher values for minimum values. Location and time are similar.
- SPI shows a higher values in moist and drought conditions than SPEI.
- Severity classification changes during drought conditions for SPEI computed by ANN for 1–month duration, but the class remains the same for 3, 6, 9, and 12–months duration.
- Lag is seen in moist and drought conditions in SPEI computed using ANN.

The limitation of these studies involves the ANN (a black box) which is still unlikely to ascertain how the performance would be for future extreme climatic data. However, a replacement with various new approaches to machine learning could enhance the existing studies. In the future, various other aspects, such as using different ET empirical techniques, neural network architecture and algorithm, could be focused on indicating their performance optimization and improvement. Various AI models evaluated PET, which is important, but their application and utilization in various other indices is still required. Further discussion about the performance of each of them when applied could lead to the development of more robust algorithms.

**Supplementary Materials:** The following supporting information can be downloaded at: https://www.mdpi.com/article/10.3390/eng4010003/s1.

**Author Contributions:** S.C.D. implemented the research work including drafting, analyzing and computation of the research manuscript under the supervision and guidance of M.S.S. and S.A. All authors have read and agreed to the published version of the manuscript.

**Funding:** This research received no external funding.

**Institutional Review Board Statement:** Not applicable.

**Informed Consent Statement:** Not applicable.

**Data Availability Statement:** Not applicable.

**Conflicts of Interest:** The authors declare no conflict of interest.

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
