# Peer review of "Assessment of Artificial Neural Network through Drought Indices"

_2673-4117, doi:10.3390/eng4010003_

Round 1

Reviewer 1 Report

The paper ‘Assessment of Artificial Neural Network through Drought Indices’ provides an interesting and innovative study on the performance of reference or potential evapotranspiration calculated using the artificial neural network model when integrated with drought indices. I do not have any major revisions for the paper, so my recommendation was for 'Accept after minor revision'.

Additional comments

1) The results and discussion section is incomplete. I only saw results and there is no discussion. Please complete this section with a discussion of your results and comparison with other reference papers already developed.

Author Response

Dear Reviewer,

Thank you very much for the requisite time and your comments. Point-by-point response has been taken into consideration and the response could be evaluated through the attachment and manuscript.

Looking forward to your further comments. 

Kind regards
Smit Doshi

Reviewer 2 Report

Although the subject of the article is interesting, there are significant problems in conveying its focus, interpreting it, and highlighting its novelty. The article should be modified due to the following reasons:

1. Line 10: The authors state that potential evapotranspiration is ETo. This leads to the complexity of the meaning. ETo or reference crop evapotranspiration refers to evapotranspiration from a reference surface not short of water. For details, see FAO Irrigation and drainage paper 56 (https://www.fao.org/3/x0490e/x0490e00.htm).

2. Line 14: What is the unit of RMSE (mm)?

3. Authors need to revise the abstract to include the research problem, objective, main results, and research recommendations that are practically achievable. More numerical results should be included in the abstract.

4. Keywords should not overlap with the title 

5. Line 92: Evapotranspiration is donated ET. However, the term evapotranspiration was used earlier. Please use the abbreviation the first time it appears.

6. Use more new references for the literature review. There should be an updated and complete literature review. The organization of the introduction section should be significantly improved. The literature review could be more systematic.

7. Figure 1 is not clear.

8. The descriptive statistics of the data (training, testing and validation) should be provided.

9. What are the hyperparameters of the ANN?

10. Which ANN model was the best estimate of ETo?

11. What are the neuron numbers of the ANN? Did you use one or more hidden layers?

12. More comparative interpretation could be expected in the paper. In the results and discussion section similarities and differences between previous studies should be discussed and the reasons for them should be pointed out. Regrettably, I could not find these discussions in the article.

13. In the Conclusions section, the authors should be objective and specific and present the main findings of the study.

14. What are the limitations and future directions of this study? This issue should be highlighted somewhere in the text of the manuscript.

15. The text contains some occasional grammatical problems. To improve readability, someone fluent in English should take care of this.

Author Response

Dear Reviewer,

Thank you very much for the requisite time and your comments. Point-by-point response to your comments has been attached.

Looking forward to your further comments.

Kind regards
Smit Doshi

Round 2

Reviewer 2 Report

The authors have greatly improved the article. The article is currently acceptable